# Cyclic Response of Steel Fiber Reinforced Concrete Slender Beams: An Experimental Study

**DOI:** 10.3390/ma12091398

**Published:** 2019-04-29

**Authors:** Constantin E. Chalioris, Parthena-Maria K. Kosmidou, Chris G. Karayannis

**Affiliations:** Reinforced Concrete and Seismic Design of Structures Laboratory, Civil Engineering Department, School of Engineering, Democritus University of Thrace, 67100 Xanthi, Greece; pkosmido@civil.duth.gr (P.-M.K.K.); karayan@civil.duth.gr (C.G.K.)

**Keywords:** steel fiber reinforced concrete (SFRC), slender beams, cyclic loading, hysteretic response, failure mode, tests

## Abstract

Reinforced concrete (RC) beams under cyclic loading usually suffer from reduced aggregate interlock and eventually weakened concrete compression zone due to severe cracking and the brittle nature of compressive failure. On the other hand, the addition of steel fibers can reduce and delay cracking and increase the flexural/shear capacity and the ductility of RC beams. The influence of steel fibers on the response of RC beams with conventional steel reinforcements subjected to reversal loading by a four-point bending scheme was experimentally investigated. Three slender beams, each 2.5 m long with a rectangular cross-section, were constructed and tested for the purposes of this investigation; two beams using steel fibrous reinforced concrete and one with plain reinforced concrete as the reference specimen. Hook-ended steel fibers, each with a length-to-diameter ratio equal to 44 and two different volumetric proportions (1% and 3%), were added to the steel fiber reinforced concrete (SFRC) beams. Accompanying, compression, and splitting tests were also carried out to evaluate the compressive and tensile splitting strength of the used fibrous concrete mixtures. Test results concerning the hysteretic response based on the energy dissipation capabilities (also in terms of equivalent viscous damping), the damage indices, the cracking performance, and the failure of the examined beams were presented and discussed. Test results indicated that the SFRC beam demonstrated improved overall hysteretic response, increased absorbed energy capacities, enhanced cracking patterns, and altered failure character from concrete crushing to a ductile flexural one compared to the RC beam. The non-fibrous reference specimen demonstrated shear diagonal cracking failing in a brittle manner, whereas the SFRC beam with 1% steel fibers failed after concrete spalling with satisfactory ductility. The SFRC beam with 3% steel fibers exhibited an improved cyclic response, achieving a pronounced flexural behavior with significant ductility due to the ability of the fibers to transfer the developed tensile stresses across crack surfaces, preventing inclined shear cracks or concrete spalling. A report of an experimental database consisting of 39 beam specimens tested under cyclic loading was also presented in order to establish the effectiveness of steel fibers, examine the fiber content efficiency and clarify their role on the hysteretic response and the failure mode of RC structural members.

## 1. Introduction

Reinforced concrete (RC) beams with inadequate transverse reinforcement exhibit a lack of ductility and fail in a rather brittle manner due to the weak tensile resistance and the reduced deflection capacity in the presence of cracks. These weaknesses can be overwhelmed by the addition of steel fibers as shear resistance mass reinforcement, which can reduce and delay cracking and ameliorate the overall performance of RC members. It is known that concrete reinforced with discrete, short, and randomly distributed fibers is a composite material with considerable cracking resistance due to the ability of the fibers to transfer the developed tensile stresses across crack surfaces (crack-bridging). The properties and the structural behavior of steel fiber reinforced concrete (SFRC) under compression, tension, flexure, shear, and torsion have been studied usually under monotonic tests, as next presented in the following subsections.

### 1.1. Compression and Tensile Behavior of Steel Fiber Reinforced Concrete (SFRC)

SFRC under compression exhibits increased strength only in mixtures with a high amount and adequate aspect ratio of fibers, whereas it exhibits a rather marginal contribution of the fibers on the compressive strength in most of the examined cases [1,2,3,4,5,6]. Nevertheless, significant improvement of the post-cracking stress–strain compressive behavior with noticeable toughness and a ductile response even in low volumetric proportions of fibers has been revealed [7,8,9,10]. A significant increase in the SFRC compressive strength is achieved in mixtures with at least 3% volume fraction of steel fibers [11,12].

The flexural tensile behavior of SFRC has been studied by using three- or four-point bending tests of small-scaled notched prismatic specimens. The majority of the experimental studies pointed out the favorable influence of the added steel fibers on the post-cracking regime [13]. Further, fibrous concrete mixtures with a high percentage of fibers showed an enhanced overall performance and a re-hardening response after cracking [2,14,15,16]. Furthermore, long steel fibers proved more effective than the short ones to ameliorate the flexural response at large deflections in terms of strength, deformation capacity, toughness, and cracking behavior [17]. Similar concluding remarks concerning the splitting tensile strength of SFRC have also been derived from extensive data of splitting tests performed on cylinders [18,19,20] and cubes [5,15]. The favorable influence of steel fiber orientation on tensile strength increase has been highlighted through splitting and flexural tensile tests on magnetically driven concrete and mortar cube specimens [21].

The orientation distribution effect of the added steel fibers has also been examined in the post-cracking tensile behavior of SFRC through tests of specially fabricated specimens subjected to direct tension and by applied different casting methods of the fresh mixtures [22]. It has been found that distribution and orientation of fibers greatly influence the overall performance of fibrous concrete. For this purpose, X-ray computed tomography techniques have recently been developed and experimentally verified, which can evaluate the microstructural parameters along with the orientation distribution and the concentration of the short fibers added in cementitious matrices as mass reinforcement [23,24]. The results of these novel techniques also confirmed that the post-cracking tensile response of fibrous concrete significantly depends on the elastic behavior of the uncracked regions of the composite material located between the cracking areas [25,26]. Micro-cracks are initially formed, propagating into localized macro-cracks after overcoming the elastic response. Hence, although fibers practically do not affect the pre-cracking behavior, it has long been recognized that SFRC exhibits increased strength and an ameliorated response after the formation of cracks due to direct tensile loading [27].

Uniaxial tests of prismatic specimens under direct tension revealed that the addition of an adequate amount of short steel fibers mainly increase the tensile load capacity, whereas longer steel fibers enlarge the ultimate tensile deformation [28]. However, SFRC with inadequate dosage of steel fibers demonstrated a negligible increase in the tensile strength and limited improvement in the post-cracking behavior [27]. Thus, a critical volume fraction of steel fibers has been proposed in order to design high-performance fibrous concrete mixtures that achieve strain hardening under direct tension with advanced ductility and energy absorption capacity [29,30,31], such as ultra-high performance SFRC [32]. Nevertheless, dispersion of long steel fibers with higher volumetric proportions was found to be problematic [33] with regard to the fine workability properties of fibrous mixtures with short lengths [34] or even microfilament steel fibers that were 13 mm long [35].

Recent analytical studies associate the tensile performance and the fracture properties of SFRC with the fiber-to-concrete interface behavior, which can be calibrated by pullout tests [36,37]. Unified formulations and simplified analytical approaches for the simulation of the overall bond behavior of fibers embedded in concrete and the prediction of the tensile response of SFRC members after cracking have also been proposed [38,39], which have been calibrated, validated or based on various pullout tests and numerical models [40,41,42]. However, due to the various available shapes, types, materials, and dimensions of fibers, along with the different mechanical properties of the cementitious mixtures, their bond characteristics also vary. Thus, there are no widely accepted or reliable constitutive tensile models that can be applied broadly in SFRC members [43]. Consequently, an interesting alternative approach has been developed by Gribniak et al. [44], which uses an inverse technique to derive average stress–strain relationships from a wide range of experimental moment–curvature curves of flexural SFRC members.

Further, the mechanical recovery of high-performance steel fiber reinforced concrete under uniaxial tensile loading has been investigated by [45,46]. It was found that fibrous concrete mixtures subjected to direct tension present multiple micro-cracks with fine crack widths, which offers an important self-healing capability [46]. The width of the developed cracks along with the type of sand and the added fibers are the main parameters that influence the self-healing capacity by generating improved interfacial bond conditions [45]. Further, a hybrid fiber reinforcing system containing polyethylene and steel fibers showed more enhanced recovery characteristics since, even in cracks with rather large width, the bond in the crack region around the longer steel fiber was bridged, controlled, and densified by the shorter polyethylene fiber [46].

### 1.2. Reinforced Concrete (RC) Structural Members with Steel Fibers in Shear and/or Flexure

Shear-critical RC structural members usually exhibit brittle catastrophic failure due to the apparent weakness of concrete in tension, which is commonly compensated by the presence of an adequate amount of steel stirrups. The advantageous characteristics of SFRC inspired researchers to investigate the use of short fibers as mass reinforcement against shear instead of conventional transverse steel reinforcement. The full or even partial replacement of stirrups is crucial in RC joints and deep, torsional, and coupling beams, where design criteria require a high ratio of shear reinforcement that leads to the extremely short spacing of stirrups and/or to the use of cumbersome reinforcement systems such as spirals, cross inclined bars, diagonal reinforcement, etc. [47,48,49,50]. Thus, at least in the critical sections of these shear-vulnerable RC members, the use of steel fibers could lead to reduced reinforcement congestion [19,51,52,53].

Several experimental works demonstrated the feasibility of substituting a significant amount of stirrups for an adequate volume fraction of short steel fibers, achieving comparable load capacities at the first shear cracking and at the maximum shear strength in RC beams [54,55,56,57,58]. Smarzewski [59,60] examined the synergetic positive effect of steel and polypropylene fibers in deep RC beams in order to replace conventional steel reinforcement. Zhao et al. [61] highlighted the favorable influence of steel fibers on the stiffness, the shear capacity, and the deformation of shear-critical RC beams, along with the reduction in crack width, height, and strains of concrete and stirrup across the diagonal section. Further, Chalioris [62] proposed an efficient analytical method to evaluate the proper dosage and type of steel fibers that should be added to concrete to replace a desirable number of stirrups and to satisfy pre-set strength and ductility requirements in shear-critical RC members. Furthermore, analytical models have also been developed to evaluate the total shear strength of concrete beams reinforced with steel fibers and longitudinal reinforcing bars, with or without stirrups [63,64,65]. Moreover, the well-known softened truss model was properly modified to implement the contribution of steel fibers on the shear strength of SFRC members [66,67].

For many years, researchers have been studying the effectiveness of steel fibers on the flexural behavior of RC structural members and it has long been acknowledged that they improved bending moment strength, ductility, failure toughness, and energy absorption capacity. Flexural cracking performance of SFRC beams with longitudinal steel reinforcing bars also appears to be enhanced with increased crack number, reduced crack width and height, restricted crack propagation, and delayed concrete spalling [68,69,70]. It has also been demonstrated that deformed or hook-ended steel fibers with higher aspect ratios can increase cracking resistance, energy dissipation, and ductility index most effectively [71,72]. Further, a recent experimental study revealed that even straight and short mill-cut steel fibers with a rather low aspect ratio equal to 40 added to a volume fraction up to 2% were capable of enhancing sectional flexural stiffness and increasing ductility related to the lateral deformations of RC columns under eccentric compression [73].

The implementation of feasible constitutive stress–strain relationships of SFRC under compression and tension in numerical analysis of concrete cross-sections reinforced with steel reinforcing bars and fibers provides rational and accurate predictions of the flexural response in terms of bending moment versus curvature analytical curves [31]. Further, the ability of steel fibers to control flexural cracking and to prevent concrete spalling inspired researchers to use SFRC in deficient and in blast-damaged RC beams that had been strengthened using externally bonded carbon fiber-reinforced polymer sheets, in order to avoid rip-off failure of the concrete cover, to prevent premature failures, to increase ductility, and to ensure structural integrity [74,75,76,77].

### 1.3. Cyclic Response of Reinforced Concrete (RC) Structural Members with Steel Fibers

The behavior of RC members under cyclic loading is significantly influenced by the fact that each concrete layer is subjected to alternate tension and compression stresses. Tests have shown that RC specimens under reversal loading suffer from degraded aggregate interlock and intensive cracking at both tension and compression zones that reduce concrete shear strength and weaken compression zones, resulting in a premature, catastrophic brittle failure [78,79,80]. The addition of steel fibers into the concrete mixture of flexural beams [72,79,81,82,83,84,85], columns [69,80]), and joints [52,86] has been proposed and examined as an alternative or additional reinforcement for seismic resistance of structures. Due to the tensile stress transfer ability of the fibers, SFRC under cyclic deformations exhibits satisfactory resistance to the formation, propagation, and widening of cracks, improved post-cracking behavior, and an ameliorated energy dissipation capability.

Daniel and Loukili [81] suggested that the use of steel fibers in flexural fibrous concrete beams with different longitudinal reinforcement ratios can be efficient to prevent an early development of macro-cracks during the pre-peak stage and enhances the energy absorption over both the elastic and inelastic stages. Tests by Campione and Mangiavillano [72] demonstrated that the addition of fibers increases the load-bearing capacity of the examined specimens, ensures more ductile behavior, and reduces degradation effects under cyclic deformations, especially in beams with increased concrete cover thickness.

Further, Harajli and Gharzeddine [82] investigated the influence of steel fibers on the bond performance of spliced steel bars in normal and high strength concrete. The experimental results showed that the existence of steel fibers delayed the formation and propagation of splitting cracks along the spliced region and increased the absorbed energy of the SFRC beams, resulting in a less brittle failure. Cyclic tests in concrete beams with high-strength steel reinforcement by Tavallali et al. [85] showed that the addition of steel fibers significantly reduced the maximum crack width of both flexural and shear cracks regarding the non-fibrous concrete beams.

Parra-Montesinos and Chompreda [83] investigated the experimental behavior of SFRC beams with and without steel transverse reinforcement. Results from the tests showed that the fibrous concrete specimens presented a pure strain-hardening behavior with advanced damage tolerance developing multiple flexural and diagonal cracks. Chalioris [84] investigated the influence of steel fibers in shear-critical RC beams with and without steel transverse reinforcement under reversal loading. Steel fibrous beams demonstrated improved overall shear performance with increased shear strength, ameliorated pre-crack and post-crack behavior, and enhanced energy dissipation capabilities compared to the non-fibrous specimens.

Cyclic experiments in a two-span continuous column performed by Kotsovos et al. [69] showed that specimens with steel fibers satisfied the performance requirements of Eurocodes for concrete strength up to 60 MPa, as the existence of steel fibers enhanced the overall performance of the specimens regarding ultimate strength and the observed failure mode. The addition of steel fibers in concrete columns with advanced high-steel longitudinal reinforcement was experimentally investigated by Lepage et al. [80]. Results from these tests showed the efficiency of steel fibers by reducing the amount of spalling of the concrete cover and altering the mode of failure from buckling of the compression bars to fracture of the tension bars.

### 1.4. Research Significance

The aforementioned literature review reveals that the majority of the conducted research is focused on the behavior of SFRC specimens under monotonic loading, whereas the cyclic response of SFRC members has been rarely investigated with regard to the plethora of the monotonically tested ones. Cyclic testing of RC beams with steel fibers is quite limited and has preliminary and exploratory character so far. Also, the hysteretic performance of such structural members is greatly influenced by several parameters which even independently have not been thoroughly clarified yet. Various combinations of conventional reinforcement (steel bars and stirrups) with steel fibers result in different contributions of the added fibers to the flexural and shear capabilities of SFRC beams. The type, the aspect ratio, and the volumetric proportion of the fibers also affect the overall structural performance.

Thus, although there are some widely accepted findings concerning the favorable influence of the added steel fibers on the post-cracking behavior of SFRC members under reversal deformations, there are still several issues that require further and systematic investigation. The dual contribution of fibers to the increase of the flexural and the shear strength of SFRC beams under different load-bearing mechanisms and the ability of steel fibers to alter the failure mode of the structural member under certain circumstances are some examples of research gaps.

Further, the recent increased interest for the application of SFRC in retrofitting applications of deficient and/or damaged RC structural members under seismic reversal excitations and the lack of relative experimental studies are the main motives behind this work.

In this work, the influence of steel fibers on the cyclic hysteretic response of slender RC beams was experimentally investigated. Three slender beams, each 2.5 m long with a rectangular cross-section, were constructed and tested under reversal deformations.

Hook-ended steel fibers, each with a length-to-diameter ratio equal to 44 and two different contents (1% and 3%), were added to the steel fiber reinforced concrete (SFRC) beams. Accompanying compression and splitting tests were also carried out in order to acquire full stress–strain relationships of the plain and the fibrous mixtures. Test results concerning the hysteretic response based on the energy dissipation capabilities, the damage indices, the cracking performance, and the failure of the examined beams are presented and discussed in later sections. Special attention is given to comprehend the observed failure modes and to associate the experimental results of this study with the test data of relevant published works from the literature. A systematic report of an experimental database consisting of 39 beam specimens tested under cyclic loading is also presented herein in order to clarify the effectiveness of steel fibers and their role on the hysteretic response and the failure mode of RC structural members.

This paper contributes to the limited existing literature on cyclic tests of RC beams with steel fibers, providing detailed experimental data of beams with a rather high amount of steel fibers (3%) that has not been examined before. An additional innovation of this paper regarding the existing experimental studies is the thorough demonstration of the available relevant tests in order to compare the experimental results and to derive new quantitative concluding remarks concerning the cyclic performance of SFRC beams.

## 2. Steel Fiber Reinforced Concrete

### 2.1. Materials

The concrete mixture used in this study consisted of a general-purpose ordinary Portland type cement (type CEM II 32.5 N, Greek type pozzolan cement containing 10% fly ash), crushed and natural river sand with a high fineness modulus, crushed stone aggregates with a maximum size of 16 mm, and water, in a mass proportion of 1:3.62:2.67:0.55, respectively. Further, 1 L retarder (Pozzolith 134 CF) per 1 m^3^ concrete was added to the mix in order to slow the rate of the concrete setting.

The steel fibers added to the fibrous concrete mixtures (see Figure 1) were hook-ended fibers to enhance the anchorage of the fiber in the matrix. The dimensions of the used fibers are an aspect ratio (length-to-diameter ratio) equal to *ℓ_f_* / *d_f_* = 44 mm/1 mm = 44, as shown in Figure 2a. Two different steel fiber volumetric proportions, *V_f_*, were chosen; 1% or 80 kg per 1 m^3^ concrete and 3% or 240 kg per 1 m^3^ concrete. The nominal yield tensile strength of the fibers was 1000 MPa.

### 2.2. Mix Preparation

The mix preparation of the steel fiber reinforced concrete (SFRC) was carried out using a pan type concrete mixer (see Figure 1). Two different SFRC mixtures were produced with steel fiber volume fractions, *V_f_*, equal to 1% and 3%, respectively. Special attention was given to the 3% SFRC mixture in order to ensure flowability of the fresh mixture and uniformity of fiber distribution, since it is known that high fiber content (≥1.5%) can decrease mechanical properties of concrete [35].

The ready-mix concrete was poured into the pan type mixer in three stages (Figure 1a–d). The steel fibers were first dispersed clump-free by hand and added steadily in small amounts into the fresh concrete mixture during stirring in order to prevent clump formation (Figure 1b,c). Stirring continued gradually, ensuring that the produced mixture would obtain uniform material consistency, adequate workability, and homogeneous fiber distribution (Figure 1d). The freshly prepared SFRC mixtures were placed in the cylinders and the molds of the specimens and adequately vibrated. During mixing and casting of the fresh mixtures, no steel fiber segregation was observed.

### 2.3. Compression and Splitting Tests

Standard concrete cylinders with a diameter to height ratio of 150/300 mm were cast from the plain concrete (PC) batch and from the batches of each SFRC mixture (*V_f_* = 1% and *V_f_* = 3%). Three specimens from each batch were tested under axial compression and three specimens under splitting tension on the day of the tests of the beams using a universal testing machine (UTM, ELE International, Leighton Buzzard, UK) with an ultimate capacity of 3000 kN. The compression tests were carried out under displacement control mode (about 2 mm/min constant rate of strain) to obtain the post-peak behavior of the SFRC mixtures. During the tests, two linear variable differential transducers (LVDTs, Kyowa, Tokyo, Japan) with 0.01 mm accuracy were installed to measure the axial strain.

The mean and standard deviation values (in parentheses) of the compressive and tensile splitting strength of the examined PC and the SFRC mixtures are given in Table 1. Figure 2b illustrates the experimental compressive behavior of the examined cases in terms of stress–strain curves. It is clear that the addition of the steel fibers to the concrete results in a low increase in the concrete compression strength, but it improves the post-peak behavior and increases the absorbed energy of the fibrous concrete. For SFRC with *V_f_* = 1%, the increase of the compression strength can be considered negligible. The slope of the descending part of the compressive stress–strain curves is increased as the fiber volume fraction increases (see Figure 2b).

## 3. Experimental Program of the Cyclic Tests

The current experimental work was carried out to investigate the hysteretic response of steel fiber reinforced concrete (SFRC) beams with conventional steel reinforcements subjected to cyclic four-point bending load. The influence of two different dosages of steel fibers (*V_f_* = 1% and 3%) was also studied.

### 3.1. Specimens’ Characteristics

The experimental program included three (3) slender beams, each 2.5 m long, tested in cyclic loading. One beam was constructed using plain concrete as the reference beam (non-fibrous specimen denoted as “B-P”) and two beams using SFRC containing 1% and 3% steel fibers (specimens “B-F1” and “B-F3”, respectively).

The geometry, the cross-sectional dimensions, and the reinforcement layout of the tested beams are presented in Figure 3. All beams had the same dimensions and conventional reinforcement. Their cross-sections have a width to height ratio of *b/h* = 200/200 mm, an effective depth of *d* = 170 mm, and a shear span of a = 1 m, with the shear span to the effective depth ratio being a/*d* = 5.9 (slender beams).

The top and the bottom longitudinal reinforcement consisted of three common deformed steel bars with a diameter of 12 mm (3∅12 top and 3∅12 bottom bars) that corresponded to a geometrical longitudinal reinforcement ratio equal to 1%. Further, the transverse shear reinforcement included closed steel stirrups with a diameter of 8 mm at a uniform spacing of *s* = 200 mm (∅8/200 mm) that corresponded to a geometrical web reinforcement ratio of 0.25%. The yield tensile strength of the bars and stirrups was *f_yl_* = 590 MPa.

### 3.2. Experimental Setup and Instrumentation

A four-point-bending experimental setup was used for the cyclic loading of the RC beams, as presented in Figure 4a. The beam specimens were simply edge-supported on roller supports 2.2 m apart in a rigid laboratory frame. The imposed load was applied using a steel spreader beam in two points 200 mm apart in the mid-span of the beams in order for the shear span to be equal to 1 m and, consequently, the span-to-depth ratio to be equal to 5.9. This way, the adopted loading scheme and apparatus simulate slender beams, although they were designed with a rather low ratio of transverse reinforcement. Thus, the design shear capacity of the reference non-fibrous beam is slightly higher than its flexural strength at yielding, but less than its ultimate flexural capacity. The specimens were subjected to increasing reversal cyclic deformation with a loading history of three loading steps with maximum deflections ±10 mm, ±25mm, and ±40 mm, respectively (Figure 4b).

The aforementioned quasi-static cyclic loading history was chosen to simulate the seismic effect and to capture the critical issues of the beams’ performance, as well as their seismic demands. Further, it is known that every excursion in the inelastic range causes cumulative damage in the structural elements. The amplitude of the inelastic excursions increases with a decrease in the period of the structural system, the rate of increase being very high for short period systems. Therefore, it is obvious that in the adopted loading program, emphasis is on the rapid increase of the rate of the inelastic excursions representing the common cases of very short period systems as the low rise buildings with dual structural systems that include RC frames and walls.

The load was imposed consistently by a pinned-end hydraulic actuator and measured by a load cell with an accuracy of 0.05 kN. The deflections of the beams were measured using five linear variable differential transducers (LVDTs, Kyowa, Tokyo, Japan) with 0.01 mm accuracy. One of the installed LVDTs was placed at the mid-span of the beams, two at a distance of 0.8 m from the supports within the left and the right shear span and two at the supports (see also Figure 4a). Load and corresponding deflection measurements were recorded continuously during the performed tests until the failure of the beams.

## 4. Test Results

The hysteretic responses of the tested specimens are presented and compared in Figure 5 in terms of the applied load versus mid-span deflection observed curves (Figure 5a) and the envelope curves of the observed maximum loads (Figure 5b). Calculated results of the flexural response of the tested beams according to the methodology proposed by Chalioris and Panagiotopoulos [31] are also presented in Figure 5b. Table 2 summarizes the test results of each beam related to the maximum cycle loads, the hysteretic energy dissipation in terms of the area enclosed within a full cycle of the load versus mid-span deflection curves, and the damage indices per each loading cycle. All specimens completed the testing protocol (three full loading cycles, see Figure 4b) and failed after the third loading cycle during the final downward loading direction. Figure 6, Figure 7, Figure 8 and Figure 9 illustrates the experimental behavior and the cracking patterns of the tested beams per each loading cycle until the total failure. During the tests, cracks that developed at the downward and upward loading directions were marked using red and green colors, respectively, as shown in the photographs of Figure 6b, Figure 7b, Figure 8b and Figure 9b.

### 4.1. Beam “B-P”

The cracking patterns per each loading cycle of the reference beam “B-P” (non-fibrous specimen) are presented in Figure 6b, Figure 7b, Figure 8b and Figure 9b. During the first loading cycle, typical vertical flexural cracks were initiated within and in the vicinity of the constant bending moment area. During the second loading cycle, new flexural cracks continued to form while the already existed cracks propagated vertically. In the upward loading direction and after the yielding of the tensional reinforcement at a load of about –55.9 kN, two flexural cracks in both shear spans developed inclined branches at their top and at their bottom ends (see Figure 7b). During the third cycle, more inclined cracks and severe flexural cracks developed without further significant increase of the applied load, as shown in Figure 8. In the upward loading direction, the critical flexural-shear crack at the right shear span continued to propagate in a stable manner.

The developed inclined cracks had shear character that decreased the concrete strength at the compression zone in both loading directions. Thus, in the final downward loading direction (after the third loading cycle, see Figure 5a and Figure 9) the beam failed, due to crushing of the weakened concrete compression zone in a rather brittle manner. The load and the mid-span deflection at that point were equal to +47.0 kN and 20.3 mm, respectively, which correspond to 71% of the maximum observed load and 51% of the maximum deflection of the third loading cycle, respectively.

### 4.2. Beam “B-F1”

Specimen “B-F1” was constructed using SFRC mixture with steel fiber volume fraction equal to 1%. Up until the second loading cycle, typical flexural vertical and thinner cracks developed within and very close to the constant bending moment area. The cracks were distributed uniformly along this region of the beam in both loading directions in comparison with the reference non-fibrous concrete beam “B-P”, as shown in Figure 6b and Figure 7b. After yielding of the tensional reinforcements at loads of about +62.0 kN and –58.0 kN, one severe flexural crack under the right point of the applied load propagated vertically and grew wider. The existence of steel fibers restrained the crack widening and splitting cracks developed at the level of the longitudinal bars located at the constant bending moment area due to the excessive vertical tensile stresses (Figure 8b).

At the final downward loading direction after the third loading cycle (see Figure 5a and Figure 9), the concrete at the tension zone was severely cracked and degraded by tensile stresses and the beam failed after concrete spalling, demonstrating satisfactory ductility. The load and the mid-span deflections at that point were equal to +72.0 kN and 70.0 mm, respectively. The maximum observed applied load was equal to +74.0 kN at a mid-span deflection of about 57.0 mm.

### 4.3. Beam “B-F3”

Specimen “B-F3” was constructed using SFRC mixture with steel fiber volume fraction equal to 3%. Up until the second loading cycle. a greater number of vertical, thinner, uniformly distributed flexural cracks were developed within and very close to the constant bending moment area in comparison with the fibrous concrete beam “B-F1”, as shown in Figure 6b and Figure 7b. Due to the higher volume fraction of the steel fibers, the number of cracks increased while the crack spacing decreased. After yielding of the tensional reinforcements at loads of about +65.5 kN and –60.8 kN, no severe flexural cracks were observed along the flexure-dominated zone of the beam due to the favorable contribution of the fibers. During the third loading cycle, one severe flexural crack under the right point of the applied load propagated vertically and grew wider. The beam exhibited a flexural response at the final downward loading direction (after the third loading cycle, see Figure 5a and Figure 9) and noteworthy ductility until failure. The load and the mid-span deflections at that point were equal to +79.0 kN and 70.0 mm, respectively.

It should be mentioned that the existence of high steel fiber volume fraction enhanced the post-peak behavior of the beam and no inclined cracks or concrete spalling were observed along the length of the beam.

## 5. Comparisons and Discussion of Test Results

To enable a better understanding of the behavioral characteristics of the fibrous concrete specimens, data in terms of the hysteretic energy absorption, the damage index, and the equivalent viscous damping (Figure 10 and Figure 11) of the tested specimens were acquired and examined in comparison with the data of the reference non-fibrous specimen.

The value of the damage index in Figure 10b was calculated according to the Park and Ang model [87]. This model is based on the idea that the seismic structural damage is expressed as a linear combination of the damage caused by excessive deformation and the damage accumulated by a repeated cyclic loading effect. Thus, the value of the damage index per loading cycle is given by the following sum:(1)$$D=\frac{\delta_{M}}{\delta_{u}}+\frac{\beta }{Q_{y}\delta_{u}}\int dE,$$
where *δ_M_* is the maximum deformation under an earthquake; *δ_u_* is the ultimate deformation under monotonic loading; *Q_y_* is the calculated yield strength; *dE* is the incremental absorbed hysteretic energy; and *β* is a non-negative parameter representing the effect of cyclic loading on structural damage given by the following expression:(2)$$\beta =\left(-0.447+0.073\frac{a}{d}+0.24n_{o}+0.314\rho_{\ell }\right)0.7^{\rho_{w}},$$
where a/*d* is the shear span to depth ratio; *n_o_* is the normalized axial stress (replaced by 0.2 if *n_o_* < 0.2); *ρ_ℓ_* is the longitudinal steel reinforcement ratio as a percentage (replaced by 0.75% if *ρ_ℓ_* < 0.75%); and *ρ_w_* is the confinement ratio.

From the observed load versus mid-span deflection envelopes of the tested specimens in Figure 5b, it can be deduced that the addition of steel fibers improves the overall behavior of the specimens “B-F1” and “B-F3” with regard to the non-fibrous reference beam “B-P”. Due to the existence of steel fibers, the first cracking load increased (see Table 2), confirming the efficiency of the fibers to delay the propagation of the macro-cracks. Further, the addition of the fibers prevented the formation of inclined cracks along the length of the fibrous specimens in both loading directions by increasing the tensile strength of concrete and the shear capacity of the beam and, consequently, by altering the brittle failure to a ductile flexural failure. Further increase of the amount of steel fibers from 1% to 3% resulted in a noteworthy ductility until failure (see Figure 5, Figure 6, Figure 7, Figure 8 and Figure 9 and Table 2).

The lower amount of the steel fibers used in the “B-F1” beam seemed to be inefficient in bridging the tensile cracks developed at the level of the longitudinal reinforcing bars, due to the degradation of the steel–concrete bond performance and the beam failure due to concrete spalling. Nevertheless, the addition of a higher volume fraction of steel fibers in “B-F3” prevented the propagation of horizontal cracking, resulting in a typical flexural failure (see also Figure 9).

The deterioration of the steel–concrete bond performance in RC members under cyclic loading was first investigated and discussed by Popov [88]. When reverse load was applied, internal tensile cracks developed around the reinforcing bars and at both sides of their ribs. During the repeated application of the cyclic load, these cracks opened and closed according to the loading direction. At a higher level of the applied load, a gap between the ribs of the steel bars and the surrounding concrete formed and the bars began to move back and forth through a small distance rather freely, which revealed the deterioration of the steel–concrete bond. The existence of the steel fibers improved the steel–concrete bond performance, as also highlighted by Daniel and Loukili [81] and Hameed et al. [79].

Figure 10a shows the absorbed hysteretic energy per loading cycle in terms of the area enclosed within a full cycle of the load versus mid-span deflection curves of the tested beams in Figure 5a. From the improved ductility of the fibrous concrete beams, it is clear that “B-F1” and “B-F3” demonstrated a higher energy dissipation capacity compared to the non-fibrous beam “B-P”, as also shown in Figure 10a and Table 2. This is also confirmed by the values of the equivalent viscous damping in Figure 11.

The calculated values of damage indices according to Park and Ang [87] are presented and compared in Figure 10b and Table 2 for the tested specimens. Values higher than 1.0 are related to the collapse of the system corresponding to the situation when the structure is no longer able to absorb additional energy. From these results it can be deduced that the fibrous beams present lower damage index factors compared to the reference beam “B-P”. Beam “B-F3”, containing a higher percentage of steel fibers, exhibited a considerable lower damage level regarding the reference beam even from the first loading cycle.

## 6. Experimental Database and Comparisons

There are only a few experimental works available in the literature investigating the hysteretic behavior of RC beams with steel fibers under cyclic loading. The parameters examined so far have been the volume fraction of the steel fibers, with a maximum amount of 1.5%, the thickness of the concrete cover, the concrete strength, and the ratio and the strength of the longitudinal reinforcement. The experimental database presented in Table 3 consists of 39 non-fibrous and fibrous RC beams subjected to reversal deformations derived from the present study and five existing works from the literature [72,79,81,82,85].

Table 3 presents the geometrical, the mechanical, and the reinforcement data and compares them with the experimental results yielded from the examined beams. The values of *ΔP_max_* express the increase of the ultimate load capacity, *P_max_*, due to the addition of steel fibers. From these values it is deduced that, although there is an obvious tendency of the SFRC beams to demonstrate increased strength with respect to the reference specimens without fibers, this increase is not consistent, neither is it analogous to the increase of the steel fiber content. This aspect was more or less expected since the provided longitudinal reinforcing bars greatly influence flexural strength and absorbed energy at a more significant level than the added steel fibers do. However, the favorable contribution of the steel fibers is focused on the increase of the shear capacity of the beams, which is higher than the corresponding increase of the flexural strength and, under certain circumstances, causes an important modification of the failure mode from brittle-shear to flexural-ductile.

Further, fibrous beams present higher values of *d_Pmax_*, which defines the displacement of the beam at the maximum load capacity, than the corresponding non-fibrous beams. This is attributed to the ability of steel fibers to increase the residual tensile stresses after cracking [7,44,89] and to delay cracking and, consequently, increase the deformation capability of the beams, even in reversal loading conditions.

Furthermore, according to the experimental results presented in Table 3, it can also be concluded that the thickness of concrete cover influences the efficiency of the steel fibers and the nature of the observed failure mode. The majority of the non-fibrous beams failed in a rather brittle manner dominated by shear failure or by concrete crushing or failed due to concrete spalling. The failure mode of the fibrous specimens with sufficient thickness of concrete cover and adequate steel fiber volume fraction (greater than 1.5%, which could be considered as a recommended lower content level of fibers) is flexural with satisfactory ductility. Otherwise, the beams tend to fail due to concrete spalling. It is also noted that, in the majority of the examined cases, the addition of steel fibers in a dosage equal to or less than 1% seems to be insufficient to improve the failure mode of the SFRC beams and to alter it from brittle to ductile. Further, the enhanced flexural behavior with noticeable ductility of concrete beams reinforced with bars and steel fibers is also attributed to the ability of steel fibers to significantly increase tension-stiffening stresses [90].

An additional conclusion that can also be deduced from the comparison of the test results of Table 3 is that the usage of steel fibers could be more efficient by taking extra care to choose the right thickness of concrete cover with the appropriate combination of a rather high steel fiber volumetric proportion.

## 7. Concluding Remarks

The experimental investigation has been conducted in this paper to investigate the influence of steel fibers on the cyclic hysteretic response of slender RC beams. The following conclusions can be drawn within the scope of this study:Based on the hysteretic responses, the cracking, and the failure modes of the tested beams, it can be deduced that the overall performance of the RC beams with steel fibers was improved with respect to the behavior of the reference specimen without fibers, confirming most of the known aspects.The non-fibrous reference specimen demonstrated shear diagonal cracking failing in a rather brittle manner, whereas the SFRC beam with 1% steel fibers failed after concrete spalling with satisfactory ductility. Moreover, it is stressed that the SFRC beam with 3% steel fibers exhibited an improved cyclic response, since no inclined cracks of shear nature or concrete spalling were observed along the length of the beam that failed due to flexure with significant ductility.The lower amount of the steel fibers seemed to be inefficient to bridge the tensile cracks developed at the level of the longitudinal reinforcing bars due to the degradation of the steel–concrete bond performance and, consequently, SFRC demonstrated concrete spalling. The addition of a higher volume fraction of steel fibers (3%) prevented the propagation of horizontal cracking, resulting in a pronounced flexural failure with enhanced post-peak hysteretic behavior in terms of strength, ductility, cracking performance, absorbed energy capability, and equivalent viscous damping.The obtained increase of the first cracking load due to the addition of steel fibers in dosage 1% and 3% was found to be 25% and 47%, respectively. Further, although the increase of the ultimate load during the third cycle of the SFRC beams with 1% and 3% fibers was only 4% and 8%, respectively (downward loading direction), and 13% and 21%, respectively (upward loading direction), compared to the reference beam, the increase of the load-bearing capacity at failure was 61% and 72% for the SFRC beams with 1% and 3% steel fibers, respectively.Based on the calculated values of damage indices, it can be deduced that the fibrous beams presented lower damage index factors than the corresponding non-fibrous reference specimen. Further, the SFRC beam containing 3% steel fibers exhibited a considerable lower damage level regarding the reference specimen even from the first loading cycle.A systematic report of an experimental database consisting of 39 beams tested under cyclic loading was also presented in order to clarify the effectiveness of steel fibers and their role on the hysteretic response and the failure mode of RC structural members. Although the amount of the examined specimens was rather limited to derive sound conclusions, it was found that the favorable influence of steel fibers on the overall performance can be achieved by taking extra care to choose the right thickness of concrete cover with a rather high amount of steel fibers (1.5% seems an appropriate lower volume content level of steel fibers).

## Figures and Tables

**Figure 1 materials-12-01398-f001:**
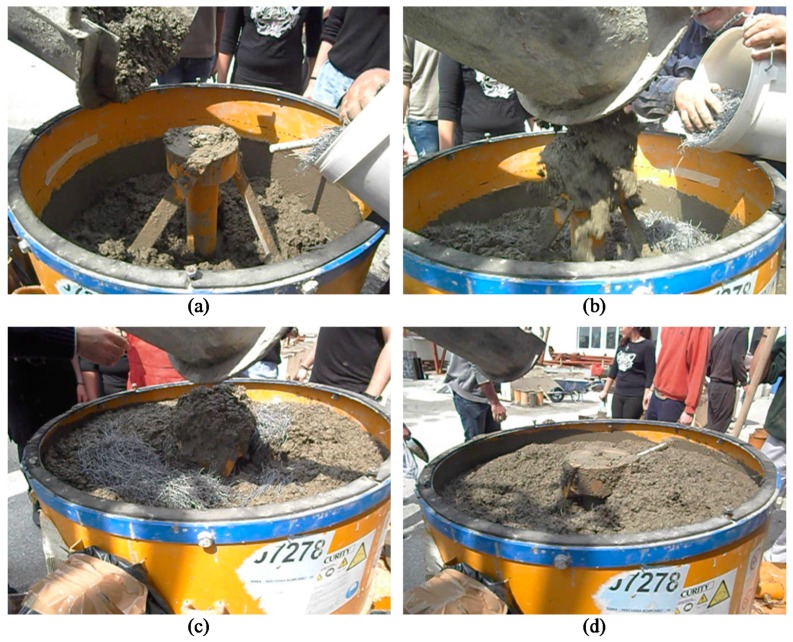
Preparation stages of steel fiber reinforced concrete: (**a**) Pouring of ready-mix concrete gradually into the pan type mixer; (**b**) and (**c**) gradual addition of the steel fibers into the fresh concrete mixture during stirring; (**d**) final homogeneous wet steel fiber reinforced concrete mixture.

**Figure 2 materials-12-01398-f002:**
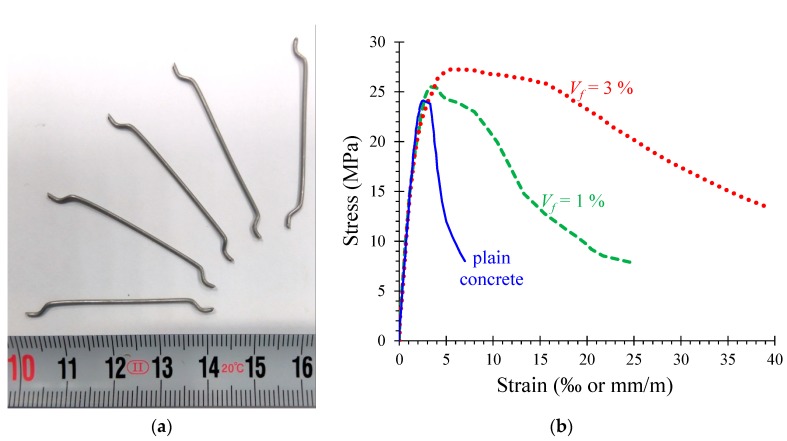
(**a**) Dimensions of the used hook-ended steel fibers; (**b**) experimental compressive stress–strain relationships of plain and steel fibrous concrete.

**Figure 3 materials-12-01398-f003:**
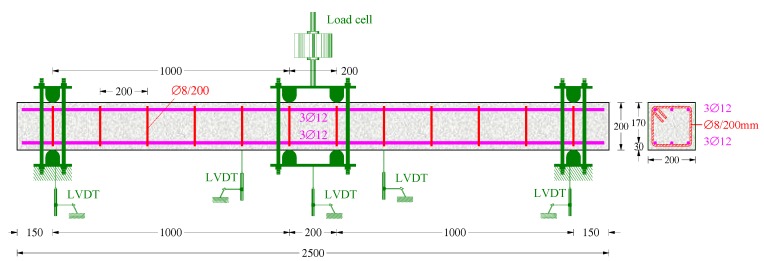
Geometry and reinforcement layout of the tested beams (dimensions in mm).

**Figure 4 materials-12-01398-f004:**
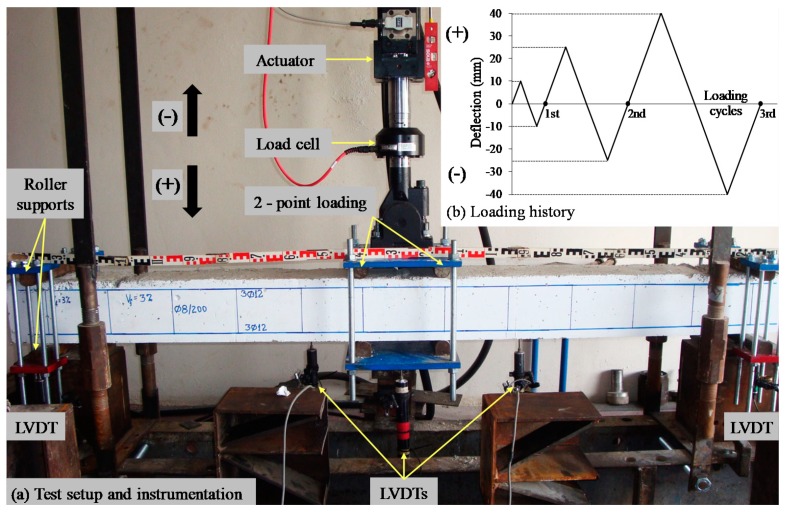
(**a**) Experimental setup and instrumentation; (**b**) loading sequence.

**Figure 5 materials-12-01398-f005:**
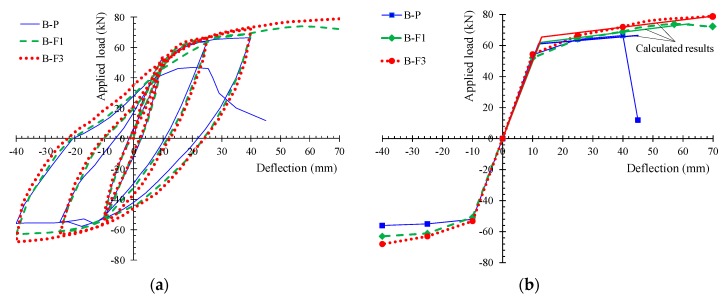
(**a**) Experimental hysteretic responses of the tested beams subjected to cyclic loading; (**b**) envelope curves of the observed maximum loads of the hysteretic response of the tested beams.

**Figure 6 materials-12-01398-f006:**
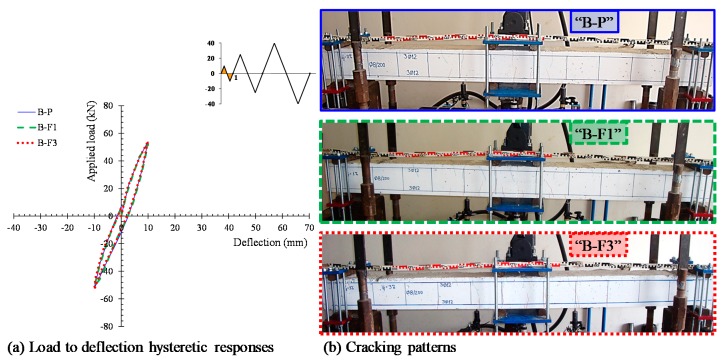
(**a**) Load to mid-span deflection observed curves and (**b**) cracking patterns of the tested beams at the end of the first loading cycle.

**Figure 7 materials-12-01398-f007:**
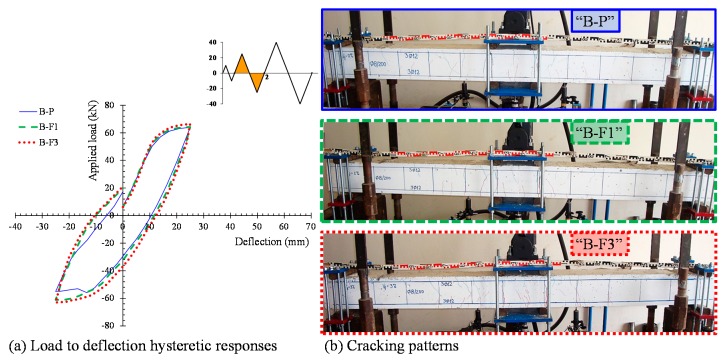
(**a**) Load to mid-span deflection observed curves and (**b**) cracking patterns of the tested beams at the end of the second loading cycle.

**Figure 8 materials-12-01398-f008:**
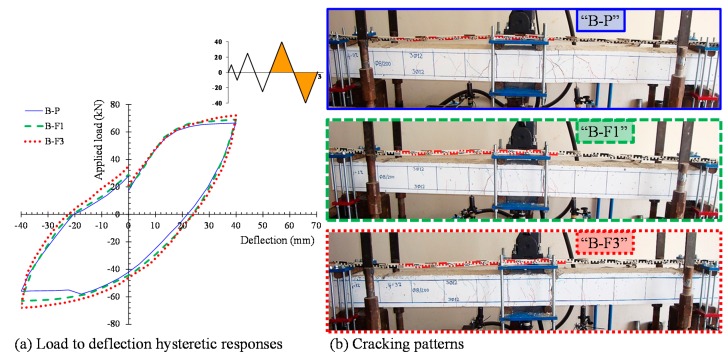
(**a**) Load to mid-span deflection observed curves and (**b**) cracking patterns of the tested beams at the end of the third loading cycle.

**Figure 9 materials-12-01398-f009:**
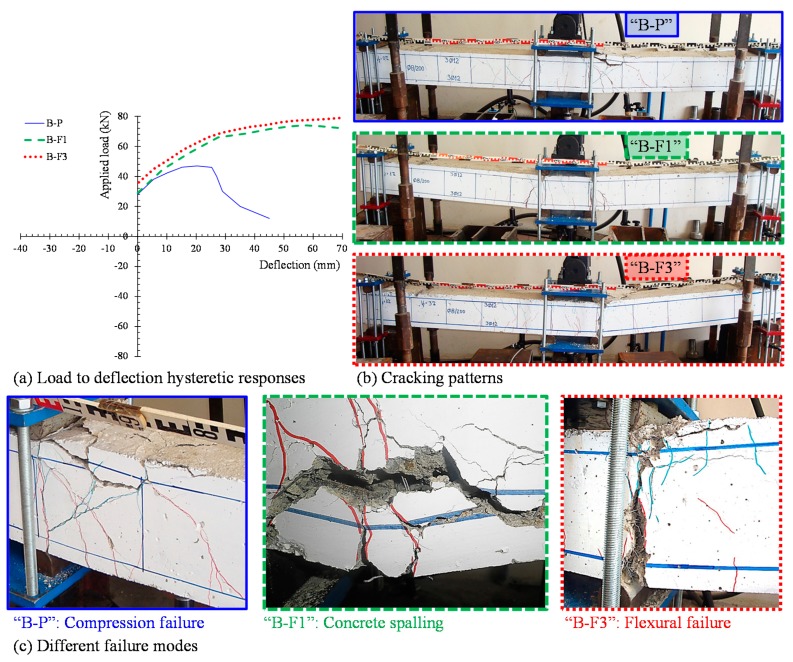
(**a**) Load to mid-span deflection observed curves, (**b**) cracking patterns, and (**c**) failure modes of the tested beams.

**Figure 10 materials-12-01398-f010:**
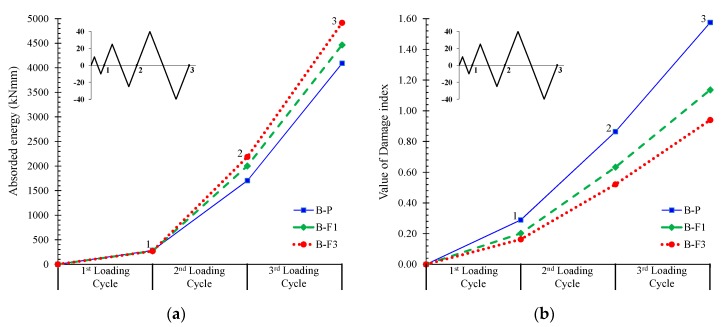
(**a**) Absorbed energy values per loading cycle of the tested beams; (**b**) comparisons of the damage indices according to Park and Ang.

**Figure 11 materials-12-01398-f011:**
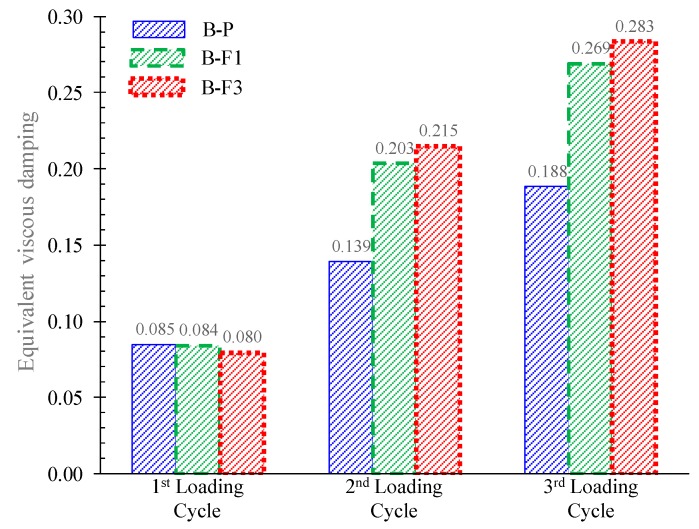
Comparisons of the equivalent viscous damping values of the tested beams.

**Table 1 materials-12-01398-t001:** Experimental results of compression and splitting tests.

Concrete Mixture	Cylinder Compressive Strength, *f_c_*	Splitting Tensile Strength, *f_ct,spl_*
	(MPa)	(MPa)
Plain concrete	24.13 (0.67)	2.62 (0.30)
SFRC with *V_f_* = 1%	25.51 (0.71)	3.32 (0.59)
SFRC with *V_f_* = 3%	27.25 (1.05)	4.96 (0.69)

**Table 2 materials-12-01398-t002:** Experimental results of the tested beams.

Cycle	Maximum Deflection (mm)	B-P			B-F1			B-F3		
		Load	Energy	Damage	Load	Energy	Damage	Load	Energy	Damage
		(kN)	(kNmm)	Index	(kN)	(kNmm)	Index	(kN)	(kNmm)	Index
1st	+10	54.0	282.5	0.29	52.0	271.4	0.20	54.4	269.1	0.16
	−10	−52.0			−50.7			−53.2		
2nd	+25	64.0	1704.5	0.86	64.2	2002.2	0.63	66.5	2186.6	0.52
	−25	−55.0			−61.1			−63.0		
3rd	+40	66.5	4091.1	1.58	69.0	4465.3	1.14	72.0	4915.9	0.94
	−40	−56.0			−63.0			−68.0		
**Cracking load:**		24.0 kN			30.0 kN			35.2 kN		
**Failure mode:**		Concrete crushing			Concrete spalling			Flexural failure		

**Table 3 materials-12-01398-t003:** Experimental data and results of beam specimens under cyclic loading derived from the literature and from the current study.

Beam Codified Name	Geometrical and Mechanical Characteristics						Reinforcements					Experimental Results			
	*b*/*h*	*d*	*c*	*L*	a/*d*	*f_c_* ^1^	*ρ_sl_*	*ρ_st_*	*L_f_*	*D_f_*	*V_f_*	*P_max_*	*d_Pmax_*	*ΔP_max_*	Failure
	(mm/mm)	(mm)	(mm)	(mm)		(MPa)	(%)	(%)	(mm)	(mm)	(%)	(kN)	(mm)		Mode ^2^
Present study															
B-P	200/200	170	16	2200	5.9	24	1.00	0.25	-	-	-	67	40	-	CC
B-F1	200/200	170	16	2200	5.9	26	1.00	0.25	44	1.00	1.0	74	57	11%	CS
B-F3	200/200	170	16	2200	5.9	27	1.00	0.25	44	1.00	3.0	79	70	19%	Fl
Daniel and Loukili [81]															
L-ref	150/300	270	16	2400	4.1	97	0.55	0.34	-	-	-	59	9.2	-	Fl
L-30	150/300	270	16	2400	4.1	110	0.55	0.34	30	0.38	1.0	86	9.9	47%	Fl
L-60	150/300	270	16	2400	4.1	116	0.55	0.34	60	0.75	1.0	93	9.9	59%	Fl
M-ref	150/300	270	14	2400	4.1	95	0.97	0.34	-	-	-	89	10.3	-	Fl
M-30	150/300	270	14	2400	4.1	112	0.97	0.34	30	0.38	1.0	116	11.0	30%	Fl
M-60	150/300	270	14	2400	4.1	117	0.97	0.34	60	0.75	1.0	124	12.1	40%	Fl
H-ref	150/300	270	12	2400	4.1	94	1.52	0.34	-	-	-	151	15.1	-	Fl
H-30	150/300	270	12	2400	4.1	114	1.52	0.34	30	0.38	1.0	159	13.7	5%	Fl
H-60	150/300	270	12	2400	4.1	117	1.52	0.34	60	0.75	1.0	179	15.0	18%	Fl
Harajli and Gharzeddine [82]															
NB20F0.0	240/300	250	40	2000	2.8	43	1.05	0.65	-	-	-	176	5.8	-	CS
NB20F0.5	240/300	250	40	2000	2.8		1.05	0.65	30	0.50	0.5	220	7.0	25%	CS
NB20F1.0	240/300	250	40	2000	2.8		1.05	0.65	30	0.50	1.0	227	12.5	29%	Fl
NB20F1.5	240/300	250	40	2000	2.8		1.05	0.65	30	0.50	1.5	228	15.0	30%	Fl
NB25F0.0	240/300	253	35	2000	2.8	43	1.62	0.65	-	-	-	217	5.5	-	CS
NB25F0.5	240/300	253	35	2000	2.8		1.62	0.65	30	0.50	0.5	242	6.0	12%	CS
NB25F1.0	240/300	253	35	2000	2.8		1.62	0.65	30	0.50	1.0	302	7.5	39%	CS
NB20F1.5	240/300	253	35	2000	2.8		1.62	0.65	30	0.50	1.5	255	7.0	18%	CS
HSC25F0.0	240/300	253	35	2000	2.8	68	1.62	0.65	-	-	-	244	5.5	-	CS
HSC25F0.5	240/300	253	35	2000	2.8		1.62	0.65	30	0.50	0.5	310	7.0	27%	CS
HSC25F1.0	240/300	253	35	2000	2.8		1.62	0.65	30	0.50	1.0	342	7.0	40%	CS
HSC20F1.5	240/300	253	35	2000	2.8		1.62	0.65	30	0.50	1.5	372	9.0	52%	CS
Campione and Mangiavillano [72]															
Beam I	150/150	133	5	550	2.1	31	1.13	0.75	-	-	-	95	6.0	-	Sh
Beam II	150/150	133	5	550	2.1	35	1.13	0.75	30	0.50	1.0	120	9.5	26%	CS
Beam III	150/150	123	15	550	2.2	31	1.23	0.75	-	-	-	105	4.0	-	Sh
Beam IV	150/150	123	15	550	2.2	35	1.23	0.75	30	0.50	1.0	126	9.5	20%	Fl
Beam V	150/150	113	25	550	2.4	31	1.33	0.75	-	-	-	112	9.5	-	Fl
Beam VI	150/150	113	25	550	2.4	35	1.33	0.75	30	0.50	1.0	100	9.5	–11%	Fl
Hameed et al. [79]															
Beam-cont	150/200	176	15	1000	2.8	41–45	0.21	0.38	-	-	-	24	8.0	-	CS
Beam-DF40	150/200	176	15	1000	2.8	41–45	0.21	0.38	30	0.50	0.5	29	8.0	19%	Fl
Tavallali [85]															
CC4-X	406/254	203	31	914	3.0	41	1.84	0.69	-	-	-	245	24.4	-	CC
UC4-X	406/254	203	33	914	3.0	43	1.24	0.69	-	-	-	236	16.0	-	CC
UC4-F	406/254	203	33	914	3.0	44	1.24	0.69	30	0.38	1.5	285	18.3	21%	Fl
UC2-F	406/254	203	33	914	3.0	44	1.24	0.34	30	0.38	1.5	271	24.4	15%	Fl
CC2-F	406/254	203	31	914	3.0	40	1.84	0.34	30	0.38	1.5	255	29.3	4%	Fl
CC4-X$	406/254	203	31	914	3.0	43	1.84	0.69	-	-	-	220	18.3	-	CC
UC2-F$	406/254	203	33	914	3.0	43	1.24	0.34	30	0.38	1.5	267	17.1	13%	Fl

^1^ Cylinder compressive strength for plain concrete or for steel fiber reinforced concrete (SFRC). ^2^ Failure mode notation: Fl: Flexural failure; CS: Concrete spalling; CC: Crushing of concrete compression zone; Sh: Shear failure.

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
