# Peer review of "Cyclic Response of Steel Fiber Reinforced Concrete Slender Beams: An Experimental Study"

_materials, 2019, doi:10.3390/ma12091398_

Round 1

Reviewer 1 Report

Well written paper with too long introduction, please reduce.

Why the cyclic loading had one cycle per applied displacement? Why not three to simulate the seismic effect?

Author Response

The authors really appreciate the positive comments and the valuable suggestions of the reviewer.

Concerning the introduction of the manuscript, it is true that it is rather long with regards to the introductory part of relative and typical journal articles. However, it is believed that since the presented literature review covers three separated parts (1.1. Compression and tensile behavior of SFRC, 1.2. RC structural members with steel fibers in shear or/and flexure, 1.3. Cyclic response of RC structural members with steel fibers) is not tedious and fits well to the wide experimental project of this study. It is noted that compression and splitting tests along with cyclic tests of SFRC beams failed in shear and flexure are included. Thus, we would prefer the introduction to remain as it is. Nevertheless, if the reviewer insists and the Editor agrees, we will reduce the introduction, appreciating additional suggestions concerning the specific parts that should be omitted.

Concerning the cyclic loading, we would like to thank the reviewer for this constructive remark. The following comments have been added in section “3.2. Experimental setup and Instrumentation” of the revised manuscript (lines 320-327):

The aforementioned quasi-static cyclic loading history has been chosen to simulate the seismic effect and to capture the critical issues of the beams performance as well as their seismic demands. Further, it is known that every excursion in the inelastic range causes cumulative damage in the structural elements. The amplitude of the inelastic excursions increases with a decrease in the period of the structural system, the rate of increase being very high for short period systems. Therefore it is obvious that in the adopted loading program emphasis is given in the rapid increase of the rate of the inelastic excursions representing the common cases of very short period systems as the low rise buildings with dual structural system that includes RC frames and walls.

In the revised manuscript all the revisions are also highlighted using red fonts.

Reviewer 2 Report

The article is very well presented and the topic is certainly interesting and within the scope of the journal. 

I attach a pdf with some minor comments.

Author Response

We really appreciate the positive comments and the valuable suggestions of the reviewer. The positive comments on the value of this study encourage the authors to work on this topic and to improve the qualification of the manuscript.

All the recommendations have been sincerely considered and the paper has been corrected, revised and ameliorated following the comments of the pdf file. In the revised manuscript all the revisions are also highlighted using red fonts. Further, please refer to the attach pdf file for the reply of each comment.
